# Effect of Carbonation on the Water Resistance of Steel Slag—Magnesium Oxysulfate (MOS) Cement Blends

**DOI:** 10.3390/ma13215006

**Published:** 2020-11-06

**Authors:** Zhiqi Hu, Yan Guan, Jun Chang, Wanli Bi, Tingting Zhang

**Affiliations:** 1Institute of Materials and Metallurgy, University of Science and Technology Liaoning, Anshan 114031, China; Anshanhzq@126.com (Z.H.); asbwl@126.com (W.B.); 2Faculty of Infrastructure Engineering, Dalian University of Technology, Dalian 116024, China; mlchang@dlut.edu.cn (J.C.); tingtingzhang@dlut.edu.cn (T.Z.); 3Research Institute of Keda Fengchi Magnesium Building Materials, Anshan 114031, China

**Keywords:** magnesium oxysulfate cement, steel slag, water resistance, carbonation

## Abstract

Magnesium oxysulfate (MOS) cement has the advantages of lightweightedness, high strength, and low thermal conductivity, but the utilization of MOS cement is limited due to low water resistance. This paper studied the influence of steel slag and CO_2_ treatment on the compressive strength and water resistance of MOS cement. The hydration products and microstructures were characterized by X-ray diffraction (XRD), thermogravimetric analysis–differential scanning calorimetry (TG–DSC), scanning electron spectroscopy (SEM), and Fourier transform infrared spectroscopy (FTIR). The results showed that the strength of MOS cement reached 89.7 MPa with steel slag and CO_2_ treatment; the water-resistance coefficients of the control and samples containing 10%, 20%, and 30% reached 0.91, 0.81, 1.01, and 1.08 MPa, respectively. The improvement in the strength and water resistance coefficients was because of carbonation that accelerated the hydration of C_2_S in the steel slag and formed a Ca–Mg–C amorphous substance. The carbonation products contributed to better water stability and denser matrix denser while inhibiting the hydration of MgO, which led to improving the water resistance of the sample.

## 1. Introduction

Global warming, which is induced by increased concentrations of CO_2_ emissions by human activities, has been the largest environmental threat of the 21st century and will become more severe. Portland cement, a major civil engineering material, emits more than 4 billion tons of CO_2_ annually, accounting for 5−10% of global emissions [1]. Therefore, it is urgent to find an alternative to Portland cement to achieve sustainable development and reduce the greenhouse effect [2].

Magnesium oxysulfate (MOS) cement is a green and environment-friendly civil engineering material prepared using caustic calcined magnesia and an aqueous solution of magnesium sulfate [3]. Since the calcination temperature of caustic calcined magnesia is lower than that of Portland cement (~800 vs. 1450 °C) [4], the carbon dioxide emitted by MOS cement is only 40−50% of that of Portland cement [5,6]. The MOS cement has the advantages of being light weight as well as having rapid strength development [7] and low thermal conductivity [8], and it is widely used in the light insulation board of the partition wall and fire coating [7]. The strength of MOS cement is due to the formation of the 517 phase (5Mg(OH)_2_·MgSO_4_·7H_2_O) [9], which is stable in water and hardly decomposes at a pH less than 12 [10]. To form a 517 phase and obtain better performance, the ratio of MgO to MgSO_4_ is often greater than 5 [11,12,13,14]. This is because although the majority of MgO is reactive, variations in the calcination temperature used to produce MgO from magnesite (MgCO_3_) produces some unreactive, dead burnt MgO [15]. Therefore, MgO is not fully involved in 517 formation, which results in excess MgO, causing a decrease in water resistance. Many studies have shown that the water resistance of MOS can be improved by adding corrective materials, such as citric acid, sodium citrate, tartaric acid, and phosphoric acid [11,12,13], whose function is to form a protective layer on the surface of MgO to prevent contact with water [12]. However, Deng [16] pointed out that a small amount of corrective materials is not enough to form a protective layer for MgO; as a result, its water resistance is not as good as ordinary Portland cement and therefore limits the utilization of MOS cement.

Steel slag is a by-product in the steelmaking process, accounting for about 25% of the steel output. In China, more than 100 million tons of steel slag is discharged every year, but the total utilization rate of steel slag is only 10% [17]. A large amount of steel slag accumulates and discharges, thus harming the environment [18]. The hydration reactivity of steel slag is low due to lack of alite (C_3_S) and amorphous silica [19]. Steel slag contains a large amount of CaO and MgO, accounting for their higher water absorption and volume of permeable voids (VPV) and limiting their use in concrete [20]. Fortunately, the soundness problem of steel slag can be solved by carbonation. Steel slag has a good carbonation-activity potential due to high lime (CaO) and belite (C_2_S) content, the carbonation of lime and belite is expressed as [21]:C_2_S + (2 − *x*)CO_2_ + *n*H**→**C*_x_*SH*_n_* + (2 − *x*)CaCO_3_(1)
Ca(OH)_2_ + CO_2_**→**CaCO_3_ + H_2_O(2)

The CO_2_ treatment of steel slag reaching a mechanical strength of 78 MPa through accelerated carbonation has been widely reported [22]. At present, there are many studies on carbonation of MgO-based cement, which can form nesquehonite (MgCO_3_·3H_2_O), lansfordite (MgCO_3_·5H_2_O), dypingite (4MgCO_3_·Mg(OH)_2_·5H_2_O), hydromagnesite (4MgCO_3_·Mg(OH)_2_·4H_2_O), and artinite (MgCO_3_·Mg(OH)_2_·3H_2_O) [23]. These hydrate magnesium carbonates (HMCs) are stable in water, improving the system density [24]. Some studies have also been made about the carbonation of MOS cement. Ba et al. [25] used 20% CO_2_ concentration to carbonate the MOS cement in a box, improving its toughness and reducing the porosity. Li et al. [26] obtained a better resistance by adding granulated blast-furnace slag to promote the carbonation of MOS cement.

Kuenzel et al. [27] studied the influence of hydromagnesite on MgO hydration and found that hydromagnesite reacted with MgO to form artinite, with a strength of 24.6 MPa. The reaction is as follows:3MgO + 4MgCO_3_·Mg(OH)_2_·4H_2_O + 11H_2_O**→**4(MgCO_3_·Mg(OH)_2_·3H_2_O)(3)
3Mg(OH)_2_ + 4MgCO_3_·Mg(OH)_2_·4H_2_O + 8H_2_O**→**4(MgCO_3_·Mg(OH)_2_·3H_2_O)(4)

At the same time, nesquehonite can also react with MgO to form artinite because nesquehonite is reactive and can be converted to hydromagnesite [28]. The reaction is as follows:MgO + MgCO_3_·3H_2_O + H_2_O**→**MgCO_3_·Mg(OH)_2_·3H_2_O(5)
Mg(OH)_2_ + MgCO_3_·3H_2_O**→**MgCO_3_·Mg(OH)_2_·3H_2_O(6)

This indicates that the products formed during the carbonation process can continue to react with the residual MgO in MOS cement, thereby improving the performance of MOS cement. Moreover, steel slag that is used to partially replace caustic calcined magnesia in MOS cement could reduce CO_2_ emission, and CO_2_ storage could be achieved by carbonating the harden pastes.

Since several studies were also carried out on the effect of carbonation on MOS cement and MOS cement-ground granulated blast-furnace slag blends. Nonetheless, the influence of carbonation on performances of MOS cement containing steel slag is still unknown. The present study investigated the effects of steel slag and CO_2_ treatment on the compressive strength and water resistance of MOS cement to address the low-utilization rate of steel slag and poor water resistance of MOS cement. Furthermore, typical samples were selected for the phase composition, phase changes, microstructure, and the pH of the hydration products for examination.

## 2. Materials and Methods

### 2.1. Raw Materials

Caustic calcined magnesia was prepared by calcining magnesite (Huafeng Magnesium Mineral Products Co., Ltd., Haicheng, China) at 850 °C for three hours; the reactive MgO was 65.11 wt.% [29]. Table 1 and Table 2 give the chemical compositions, physical properties and phase compositions of the caustic calcined magnesia and steel slag (Anshan Iron and Steel Group Corporation, Anshan, China). The particle size distribution that was obtained by laser diffraction diameter analyzer (BT-9300s, BETTER, Dandong, China) is shown in Figure 1. The MgSO₄·7H₂O and citric acid that were used in the experiments were obtained as analytical-grade reagents.

### 2.2. Preparation

Table 3 shows the paste mix designs used in this study. MOS cement was prepared by fixing a molar ratio of reactive −MgO to MgSO_4_ of eight. Epsomite was initially dissolved in water. The MOS cement paste was prepared by mixing MgSO_4_, steel slag, caustic calcined magnesia powder with citric acid in a blender according to JC/T 729-2005. First, the mixture was stirred at a low speed rate of 60 r/min for 120 s, then the stirring was stopped for 15 s, and then the mixture was stirred at a high rate of 300 r/min for 120 s. Mixed paste samples were poured into polypropylene molds (40 mm × 40 mm × 40 mm) and cured in a curing box at 60 ± 5% humidity and a temperature of 25 ± 2 °C before being demolded.

For air curing, the samples were demolded approximately 24 h after casting, and then cured in a curing box at 60 ± 5% humidity and a temperature of 25 ± 2 °C. For CO_2_ curing, the samples were demolded approximately 24 h after casting, then placed into the carbonation chamber at 0.5 MPa of CO_2_ pressure for 4 h. The samples, after the CO_2_ treatment, were cured in the curing box at 60 ± 5% humidity and a temperature of 25 ± 2 °C.

### 2.3. Testing Methods

For the compressive strength test, six samples of the MOS cement paste were measured using an electronic servo testing machine (DYE-300S, Cangzhou Jingwei Instrument Equipment Manufacturing Co., LTD, Cangzhou, China) at 3 days, 7 days, and 28 days with air curing, and with immersion in water at 28 days after air curing for 28 days. The maximum load of the machine was 300 kN, and the loading speed was 0.6 kN/s.

The water resistance coefficient, *R*_f_, for evaluating the water resistance of MOS cement was obtained as follows:*R*_f_ = *R*_w_/*R*_a_(7)
where *R*_w_ is the compressive strength of the samples water immersed for 28 days, and *R*_a_ is the compressive strength of the samples before being immersed in water.

The pastes were ground to pass through an 80 μm sieve for the XRD test (XRD, Malvern Instruments Limited and PANalytical B.V. Malvern, UK) with settings as follows: *λ*_Cu_ = 0.15418 nm, tube pressure: 40 kV, tube flow: 40 mA, start angle = 5°, end angle = 70°, step size = 0.13°, time per step = 5 s. The Rietveld method, as implemented in the Topas 6.0 software (Bruker, Karlsruhe, Germany), was used for the quantitative analysis of the mineral phases in the MOS cement samples by fitting the peak areas [30]. The analytical-grade reagent ZnO was mixed in the powder of the tested samples at 15% by mass.

Thermogravimetric analysis (STA 449F3, Netzsch, Selb, Germany) was used to analyze the powdered samples heated from 25 to 1100 °C at a uniform rate of 10 °C/min in an N_2_ gas flow of 50 mL/min.

Mercury intrusion porosimetry (MIP, Quantachrome Autoscan 60, Boynton Beach, FL, USA) was used to characterize the pore-size distributions of the specimen. The samples cured at 28 days and immersed in water at 28 days after air curing for 28 days were cut into 2–4 mm pieces and immersed in isopropanol for 24 h to stop hydration. Finally, the samples were dried in a vacuum drying chamber at 50 °C for 24 h prior to testing with a contact angle of 140° and a surface tension of mercury of 0.48 N/m. Scanning electron microscopy (SEM) (ZEISS SIGMA HD, Jena, Germany) was used to observe the micromorphologies of the hydration products of the Pt-coated samples.

## 3. Results and Discussion

### 3.1. Mechanical Properties of Samples with and without CO_2_ Treatment

The compressive strengths of the control, SS10, SS20, SS30, SS40, and SS60 at 3 days, 7 days, and 28 days are shown in Figure 2 and Figure 3. In this study, the compressive strength of the samples containing 10%, 20%, 30%, 40%, and 60% of steel slag at 28 days decreased by 29.9%, 51.8%, 66.6%, 90.4%, and 93.3%, respectively, compared with the control. It can be seen that the strength of the air-cured samples decreased gradually with an increase in the steel-slag content. The decline in the strength could be because the pH of the steel slag was 12–13 [31], while the pH of the MOS cement was 9.0–9.5 [11]. The addition of steel slag raises the pH of the system; furthermore, the CaO in the steel slag reacts with sulfate to form gypsum (CaSO_4_·2H_2_O), reducing the concentration of sulfate ions in the slurry that inhibits the 517 phase formation. The compressive strength improved significantly after CO_2_ treatment. The compressive strength of the samples containing 10%, 20%, 30%, 40%, and 60% of steel slag increased by 57.6%, 103.6%, 186.3%, 403.1%, and 651.8%, respectively. This is because carbonation promoted the hydration of C_2_S, leading to an increase in the strength [26]. When curing for 28 days, the compressive strength of the control with CO_2_ treatment was lower than that of air curing due to further hydration of MgO (Table 4). MgO formed Mg(OH)_2_ with a volume expansion of 147% [32], leading to expansion cracking and decline in strength. The results are shown in Section 3.2.

When the content of steel slag in samples is greater than 30 wt.%, even with CO_2_ treatment promoting the hydration of steel slag, the compressive strength is low due to the reduction of the cementing material, and it is difficult to meet the requirements of civil engineering. The following mainly discusses the control, SS10, SS20, and SS30.

Figure 4 shows the compressive strength and water-resistance coefficient of the samples after being immersed in water for 28 days. After being immersed in water for 28 days, the compressive strength of the control, SS10, SS20, and SS30 with air curing decreased from 81.2, 56.9, 39.1, and 27.1 MPa to 70.5, 42.5, 30.0, and 13.3 MPa, respectively. The water-resistance coefficient decreased with an increase in the steel-slag content. After the CO_2_ treatment, the water-resistance coefficient of the control, SS10, SS20, and SS30, reached 0.91, 0.81, 1.01, and 1.08, respectively, indicating that the CO_2_ treatment can improve the water resistance of the samples.

### 3.2. Hydration Product of Samples with and without CO_2_ Treatment

Figure 5 shows the diffraction pattern of the control, SS10, SS20, and SS30 at 28 days air curing. It also shows the most prominent peaks in the paste of the brucite, 517 phase, gypsum, magnesite, quartz, and periclase. An increase in the steel slag content decreased the formation of the 517 phase—one of the main reasons for the strength decline. Simultaneously, a reaction between the CaO in steel slag and the SO_4_^2-^ in the slurry resulted in the formation of the gypsum phase.

Figure 6 shows the diffraction pattern of the control, SS10, SS20, and SS30 with CO_2_ treatment at 28 days. Compared with the samples without CO_2_ treatment, the content of Mg(OH)_2_ in the control recorded an increase because carbonation accelerated the hydration of C_2_S [33], MOS cement, and MgO. The hydration of MgO can lead to expansion cracking, thus reducing the strength of the control. After the addition of steel slag, the peak of Mg(OH)_2_ is weaker than the samples without the CO_2_ treatment, which indicates that the CO_2_ treatment can reduce the content of Mg(OH)_2_.

Figure 7 shows the influence of steel slag on XRD patterns of the control and SS30 with and without CO_2_ treatment after water immersion for 28 days**.** After 28 days of immersion, the samples with CO_2_ treatment showed weaker peaks of Mg(OH)_2_ than the samples without CO_2_ treatment. This indicates that the CO_2_ treatment inhibited the hydration of MgO. Therefore, the water resistance of MOS cement can be improved by the steel slag and CO_2_ treatment. Additionally, no new phase formation was found in XRD, indicating that the carbonation products existed mainly in the amorphous form.

Table 4 shows the results of the Rietveld quantitative phase analysis of the control, SS10, SS20, and SS30 with CO_2_ treatment before and after immersed in water for 28 days. Figure 8a shows the Rietveld analysis plots of SS10 with CO_2_ treatment after water immersion for 28 days. After water immersion for 28 days, the content of MgO in the control decreased by 42.1%, whereas it decreased in SS10, SS20, and SS30 by 47.7%, 5.4%, and 0.5%, respectively; in contrast, the content of Mg(OH)_2_ increased by 20.4%, 95.4%, 18.3%, and 11.5%, respectively (Figure 8b). This means that the formation of Mg(OH)_2_ and the hydration of MgO was inhibited by the steel slag and CO_2_ treatment when the content of the steel slag in the samples exceeded 20%. In addition, the content of the 517 phase of SS20 and SS30 increased after immersion due to a decrease in the pH of steel slag after carbonation [26]. The content of C_2_S and C_4_AF also decreased significantly after immersion, and the reason for an increase in the compressive strength could be the hydration of C_2_S that formed C–S–H.

The FTIR spectra of the control, SS10, SS20, and SS30 with CO_2_ treatment for 28 days are shown in Figure 9. The peaks at ~890, ~1080, ~1425–1507, 1650, 3400, and 3700 cm^−1^ are due to a bend in the vibrations of CO_3_^2-^ [34], the stretching vibrations of SO_4_^2-^ [9], the asymmetric stretching vibration of CO_3_^2-^ [35], the vibration of H_2_O [13], the free O–H vibration of H_2_O [9], and the asymmetric stretching vibration of O–H [13]. It can be seen that with an increase in the steel slag content, the carbonate absorption band increased after carbonation. Previous studies have reported only a single absorption band between 1450 and 1480 cm^−1^ for magnesite; however, six absorption bands in the range of 1425–1507 cm^−1^ were found in this study, indicating six different carbonate ion environments [36], which meant a change in the structures of carbonate ion.

Figure 10 shows the thermogravimetric analysis–differential scanning calorimetry (TG–DSC) curves of the control, SS10, SS20, and SS30 with CO_2_ treatment after 28 days of water immersion. There were five processes of decomposition [9]:5Mg(OH)_2_·MgSO_4_·7H_2_O→5Mg(OH)_2_·MgSO_4_·5H_2_O + 2H_2_O ~100–120 °C
5Mg(OH)_2_·MgSO_4_·5H_2_O→5Mg(OH)_2_ + MgSO_4_ + 5H_2_O ~120–150 °C
Mg(OH)_2_→MgO + H_2_O ~350–450 °C
MgCO_3_→MgO + CO_2_ ~500–600 °C
5MgO + MgSO_4_→6MgO + SO_3_ ~950–1000 °C

As can be seen, with an increase in the steel slag and CO_2_ treatment, the dehydration and decomposition of MOS became more complex. Two endothermic peaks were observed between 500 and 600 °C, indicating two forms of magnesium carbonate; the former is caused by nondecomposition of MgCO_3_ [37] and the latter by the MgCO_3_ crystal decomposition. Thus, it can be proven that new phases were formed in the MOS cement when mixed with steel slag and treated with CO_2_. As well, a significant weight loss in the control at 990 °C and no obvious weight loss after the steel slag addition indicate that the addition of steel slag changed the structure of sulfate.

### 3.3. Microstructure of the Samples with and without CO_2_ Treatment

The SEM images of the control and SS30 after air curing for 28 days are shown in Figure 11. Since the MOS cement was difficult to carbonate, the microstructure of the MOS cement did not change before and after carbonation. A large number of needle-like whiskers in the pole of the control were observed, which are identified as the 517 phase (Mg:S:O = 6.12:1:25); the matrix was compact. With the addition of the steel slag, the 517 phase whiskers in the pores disappeared—an observation consistent with the XRD detection results. However, the structure in the matrix was relatively loose, and it contained gypsum (Ca:S:O = 0.87:1:10) and Mg(OH)_2_ (Mg:O = 1.77:1). After the CO_2_ treatment, the matrix became more compact and was filled with a large amount of Ca–Mg–C amorphous substance (0.31:1:3.51). This densified the matrix and led to its strength increase. Besides this, nesquehonite was found on the surface of the control, indicating that the CO_2_ treatment of the MOS cement can form HMC substances.

Figure 12 shows the backscattered electron image of the control and SS30 with CO_2_ treatment after water immersed for 28 days. It can be seen that a large number of cracks were generated in the control, which possibly decreased the strength of the matrix. However, with the steel slag mix and the CO_2_ treatment, the matrix maintained its original morphology. The Ca–Mg–C amorphous substance wrapped around the MgO particles in the matrix to prevent the MgO from coming into contact with water, thus inhibiting the MgO hydration and improving the water resistance of the samples.

Figure 13 shows the pore-size distributions of the control and SS10 with CO_2_ treatment before and after the immersion. After 28 days of immersion in water, the micropores of the control decreased and a small number of pores with a diameter greater than 1000 nm were formed, so the strength of the control decreased. The result is consistent with the results of backscattered electron image. The porosity of SS30 is greater than the control, leaving enough space for the hydration of MgO or the MgO and HMC substances reaction. The porosity in SS30 decreased with CO_2_ treatment after immersion; the matrix became more compact, thus increasing the compressive strength.

For a general Ca-rich substance, the carbonation process consists of a dissolution of Ca(OH)_2_, dissolution of CO_2_ to form CO_3_^2−^, and calcium carbonate precipitation [38].
Ca^2+^(aq) + CO_3_^2−^(aq)**→**CaCO_3_(s)(8)

Calcium carbonate precipitates in a supersaturated suspension. For a system with a reactive MgO, Mg^2+^ was provided by MgO and Mg(OH)_2_, and the supersaturated suspension in pores contained both Mg^2+^ and Ca^2+^, forming carbides with different Mg/Ca ratio (Equation (9)).
Ca^2+^(aq) + Mg^2+^(aq) + CO_3_^2−^(aq)**→**Ca*_x_*Mg_1−*x*_CO_3_(s)(9)

Only a minimal amount of Mg^2+^ dissolved into the suspension because of low brucite solubility. The dense carbonated layers prevented MgO and Mg(OH)_2_ from contacting air and water, thus inhibiting the MgO hydration [39]. Carbonation promoted the hydration of pastes and C_2_S, thereby increasing the strength. Furthermore, the HMC substances formed by carbonation dissolved CO_3_^2-^ and inhibited MgO hydration, forming Mg(OH)_2_ (Figure 14). It reacted with MgO to form a stable amorphous substance that filled the cracks and increased the strength after immersion in water.

## 4. Conclusions

This work investigated the use of steel slag and CO_2_ treatment for water resistance and microstructure of the MOS cement. The compressive strength, water resistance, phase composition, and microstructure of MOS cement after the addition of steel slag and CO_2_ treatment were discussed. The following conclusions can be drawn:With an increase in the steel slag content, the compressive strength and the water resistance of MOS cement decreased. The presence of CaO in steel slag increased the pH of the pastes and reacted with SO_4_^2-^ to form gypsum, which reduced the concentration of sulfate ions in slurry, and both of which were not conducive to the formation of 517 phase, so the compressive strength of the pastes decreases.The compressive strength of the samples had a significant increase after carbonating, which was mainly due to the promotion of C_2_S hydration in steel slag after carbonation.The products (Ca–Mg–C amorphous substance) of carbonation exhibited good water stability as they densified the matrix, thus leading to an improved compressive strength of the MOS cement.The HMC substances were formed by carbonation dissolved CO_3_^2-^ when immersed in water, which limited the dissolution of Mg^2+^ and inhibited MgO hydration forming Mg(OH)_2_. The HMC substances reacted with MgO to form a stable amorphous substance that filled the cracks and increased the strength after immersion in water.Pure MOS cement has low porosity. The hydration of MgO after immersion caused cracking as there was no space for Mg(OH)_2_ formation. The addition of steel slag increased the porosity of the samples, and the matrix became denser after carbonation and water immersion. Although it still had a few void regions, the average diameter of the pores decreased, enhancing the compressive strength.Using steel slag that partially replaced caustic calcined magnesia can reduce CO_2_ emissions, as an alternative to a sustainable development of Portland cement.

In conclusion, with the addition of steel slag and CO_2_ treatment, the water resistance of MOS cement was improved. Moreover, it can also be regarded as an alternative to create a sustainable concrete industry by storing CO_2_ and reducing the CO_2_ emissions.

## Figures and Tables

**Figure 1 materials-13-05006-f001:**
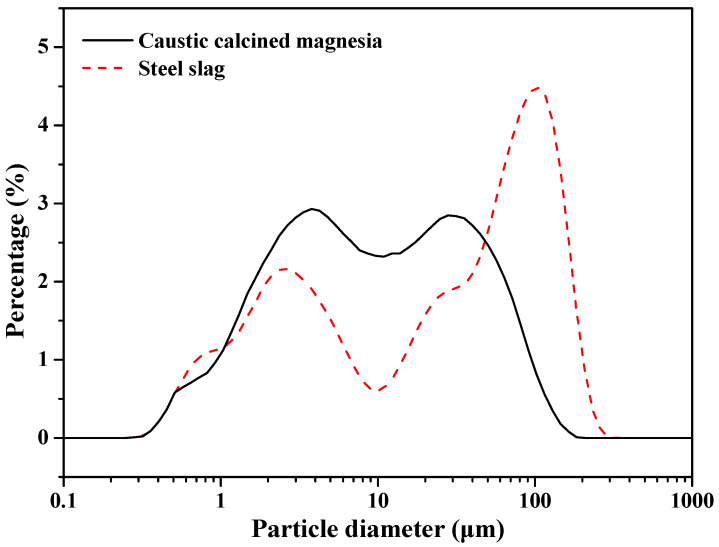
Particle size distributions of the caustic calcined magnesia and steel slag used in this research.

**Figure 2 materials-13-05006-f002:**
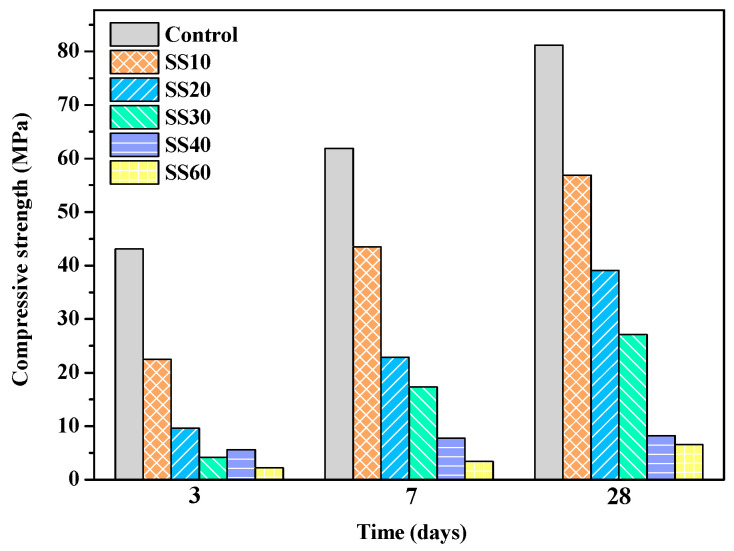
Compressive strengths of the control, SS10, SS20, SS30, SS40, and SS60 with air curing at 3 days, 7 days, and 28 days.

**Figure 3 materials-13-05006-f003:**
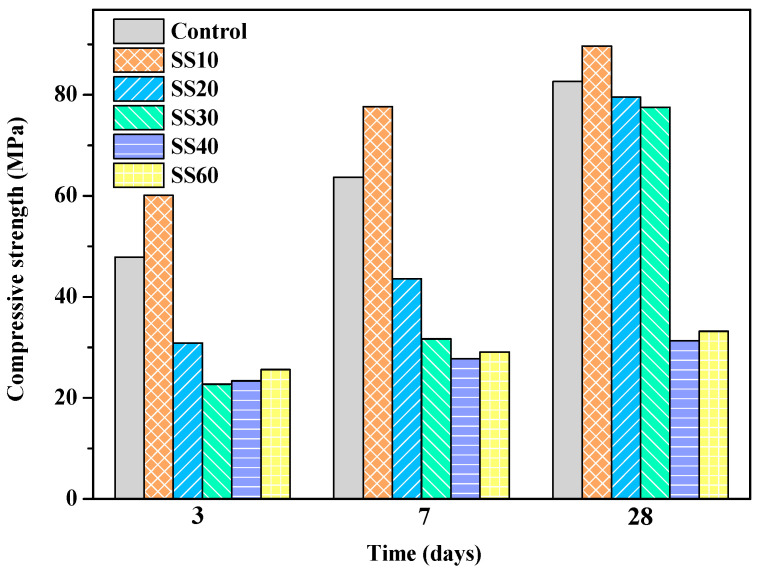
Compressive strengths of the control, SS10, SS20, SS30, SS40, and SS60 with CO_2_ treatment at 3 days, 7 days, and 28 days.

**Figure 4 materials-13-05006-f004:**
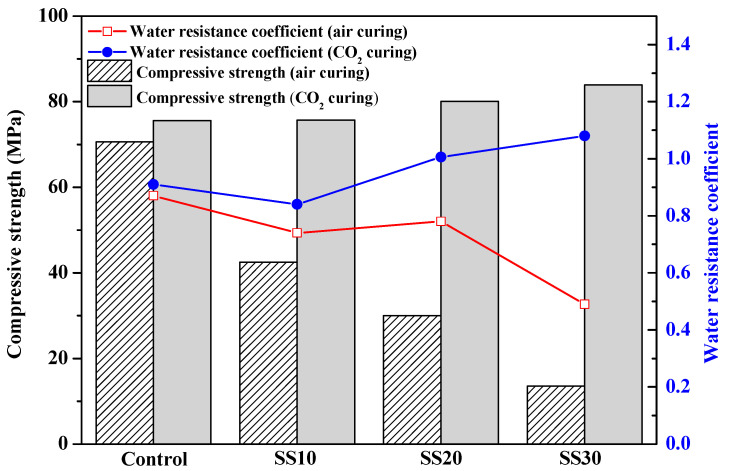
Compressive strengths and water-resistance coefficient of the control, SS10, SS20, and SS30 with and without CO_2_ treatment after water immersion for 28 days.

**Figure 5 materials-13-05006-f005:**
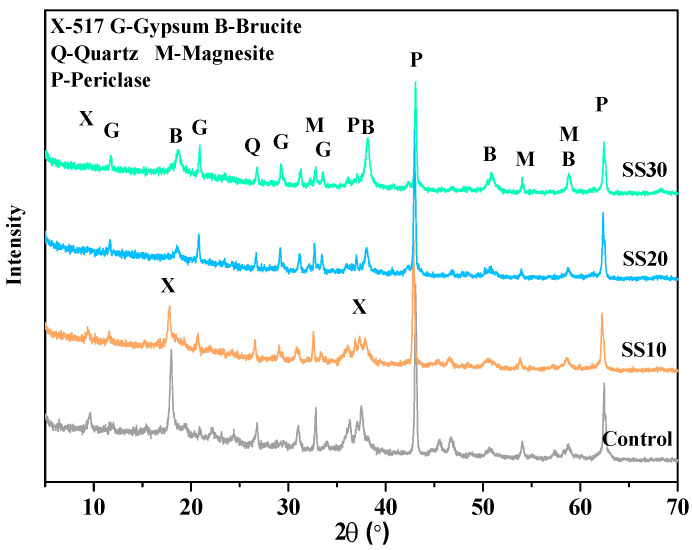
Powder XRD patterns of the control, SS10, SS20, and SS30 with air curing at 28 days.

**Figure 6 materials-13-05006-f006:**
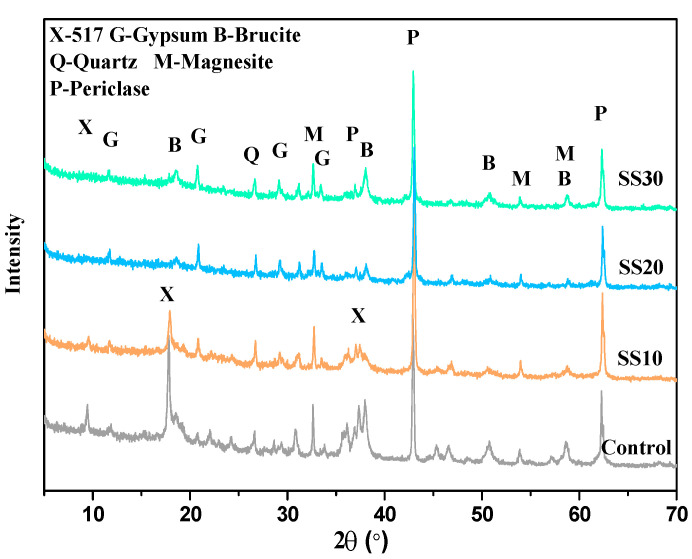
Powder XRD patterns of the control, SS10, SS20, and SS30 with CO_2_ treatment at 28 days.

**Figure 7 materials-13-05006-f007:**
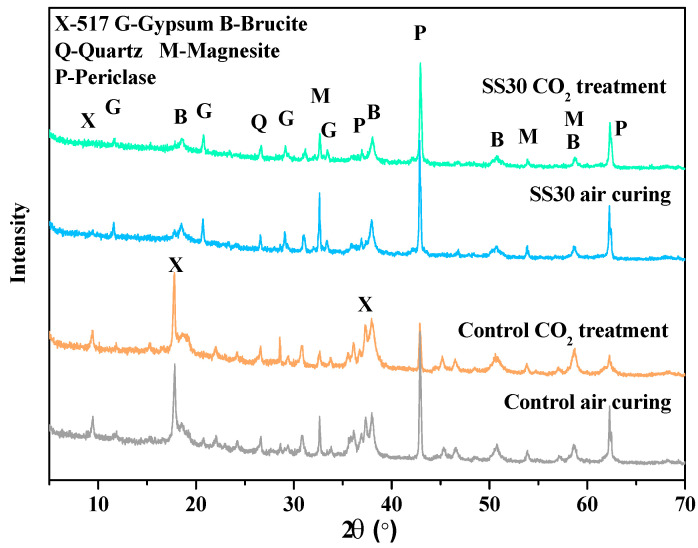
Powder XRD patterns of the control and SS30 with and without CO_2_ treatment after water immersion for 28 days.

**Figure 8 materials-13-05006-f008:**
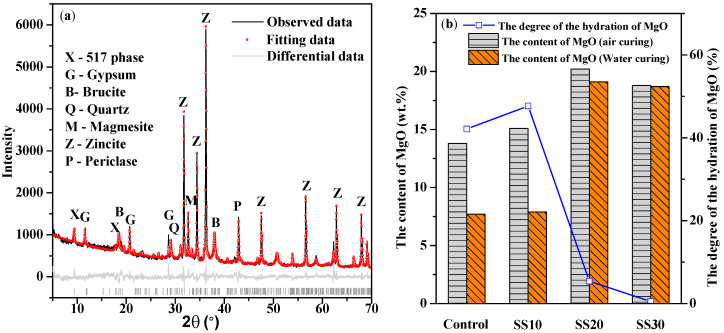
(**a**) Rietveld analysis plots of SS10 with CO_2_ treatment after water immersion for 28 days and (**b**) the content of MgO in the control, SS10, SS20, and SS30 with CO_2_ treatment before and after water immersion for 28 days.

**Figure 9 materials-13-05006-f009:**
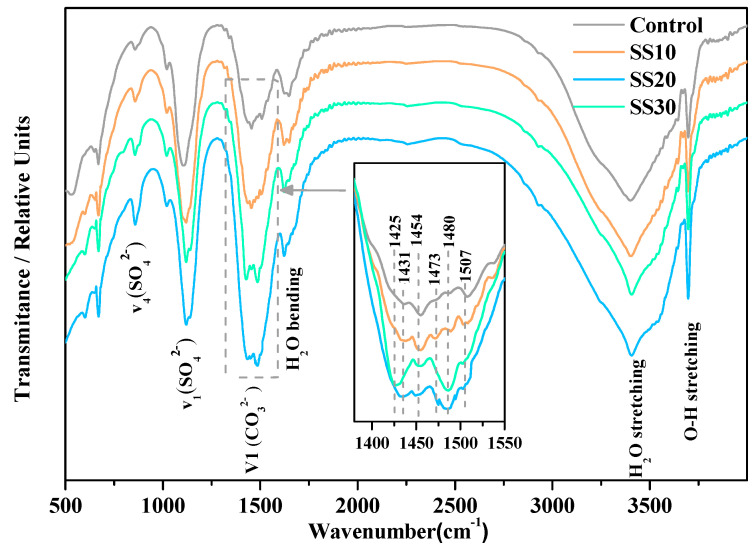
FTIR spectra of the control, SS10, SS20, and SS30 with CO_2_ treatment after 28 days curing.

**Figure 10 materials-13-05006-f010:**
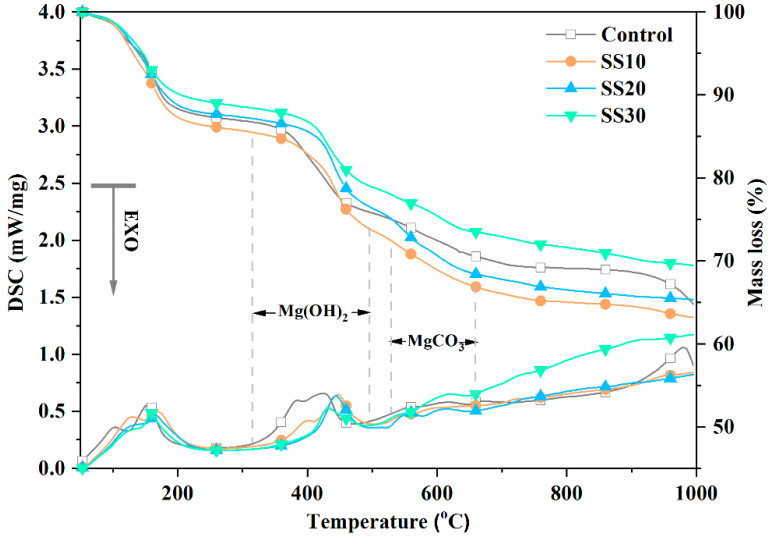
Thermogravimetric analysis–differential scanning calorimetry (TG–DSC) analysis of the decomposition temperatures of the control, SS10, SS20, and SS30 with CO_2_ treatment after 28 days curing.

**Figure 11 materials-13-05006-f011:**
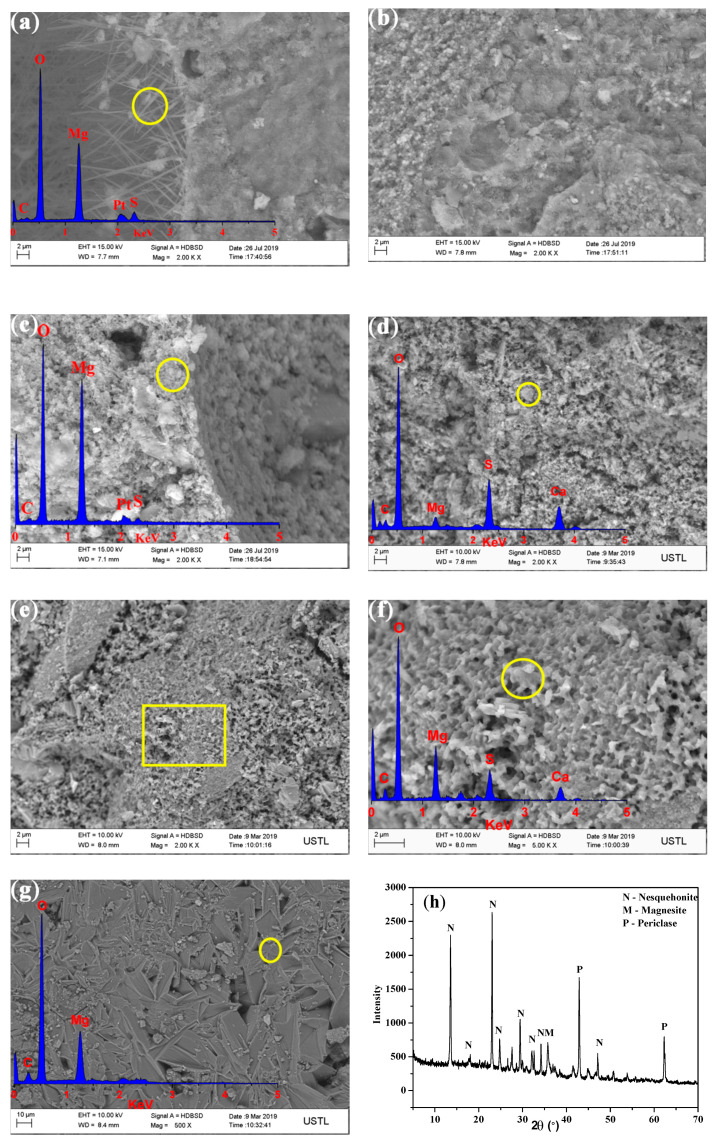
Fractured surface morphology of samples after 28 days curing: (**a**) pore of control with CO_2_ treatment; (**b**) matrix of control with CO_2_ treatment; (**c**) pore of SS30 with air curing; (**d**) matrix of SS30 with air curing; (**e**,**f**) matrix of SS30 with CO_2_ treatment; (**g**) surface of control with CO_2_ treatment; and (**h**) XRD pattern of the surface of the control with CO_2_ treatment.

**Figure 12 materials-13-05006-f012:**
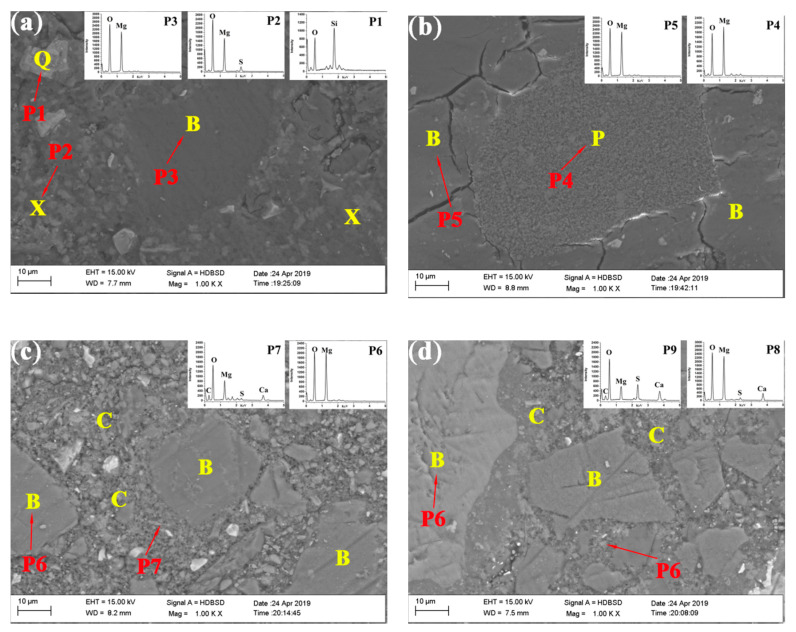
Backscattered electron image of the control and SS30 with CO_2_ treatment: (**a**) matrix of control after 28 days air curing; (**b**) matrix of control after 28 days water immersion after air curing for 28 days; (**c**) matrix of SS30 after 28 days air curing; (**d**) matrix of SS30 after 28 days water immersion after air curing for 28 days; X: 517 phase, B: Brucite, Q: Quartz, C: Ca–Mg–C amorphous substance; P: Periclase; P1 is Quartz; P2 is 517 phase; P3, P5, P6, and P8 are Brucite; P5 is Periclase; P7 and P8 are Ca–Mg–C amorphous substance).

**Figure 13 materials-13-05006-f013:**
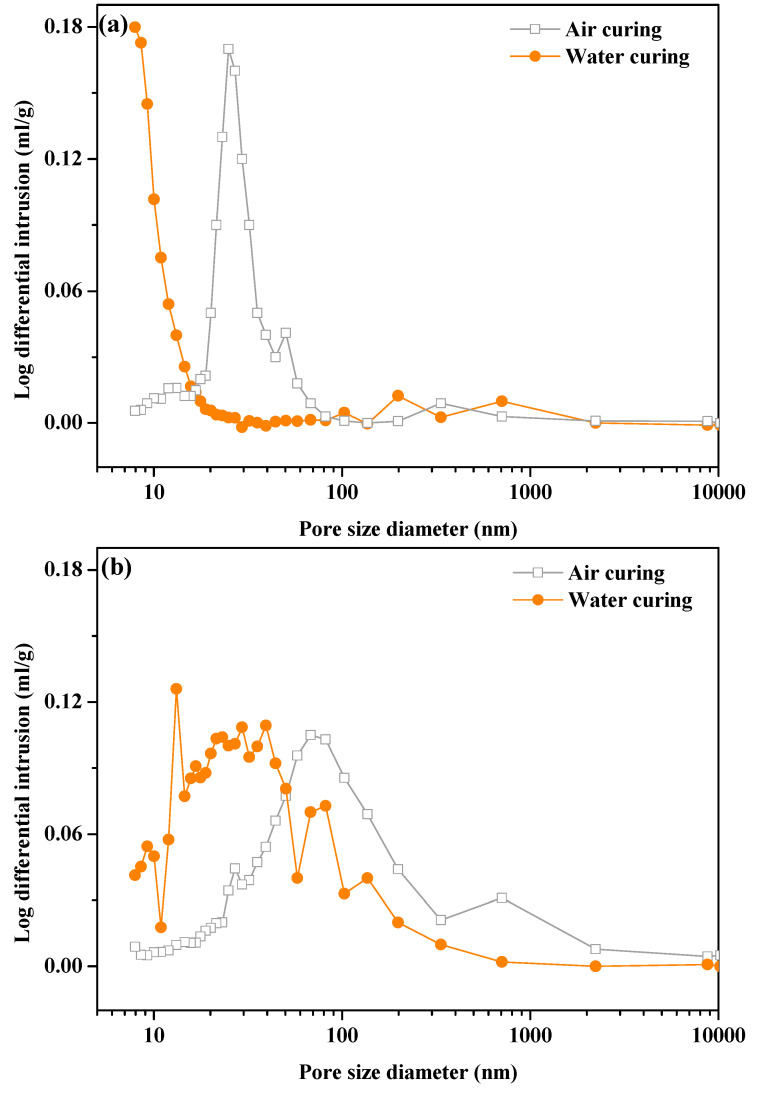
Comparison of cumulative intruded volume vs. pore diameter after 28 days of air curing and after water immersion for 28 days: (**a**) control and (**b**) SS30.

**Figure 14 materials-13-05006-f014:**
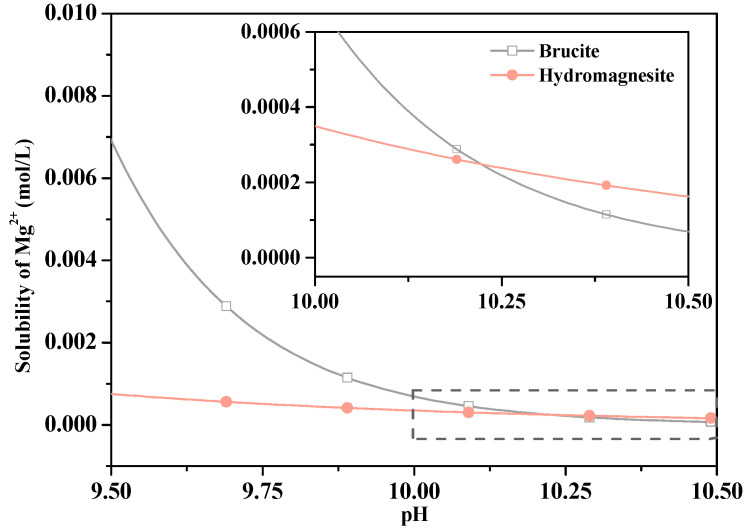
Relationship between the Mg^2+^ solubility and pH in brucite and hydromagnesite (the *K*_sp_ values of brucite and hydromagnesite are −11.16 and −38.47, respectively [40,41]).

**Table 1 materials-13-05006-t001:** Chemical compositions and physical properties of caustic calcined magnesia and steel slag.

Content (wt.%)	Caustic Calcined Magnesia	Steel Slag
MgO	85.6	9.19
CaO	2.46	37.85
Fe_2_O_3_	1.60	25.00
Al_2_O_3_	0.23	5.13
SiO_2_	5.56	18.20
LOI	4.50	1.80
Density (g/cm^3^)	2.94	3.45
Water absorption (wt.%)	20.17	4.87
Specific surface area (m^2^/g)	25.3	15.7

**Table 2 materials-13-05006-t002:** Phase compositions of caustic calcined magnesia and steel slag.

Component	Content (wt.%)								*R* _wp_
	MgO	MgCO_3_	SiO_2_	C_2_S	C_4_AF	CaO	FeO	ACn	
Caustic calcined Magnesia	80	9.9	1.6	-	-	-	-	8.5	5.766
Steel slag	5.9	-	-	27.9	17.5	3.9	2.6	42.5	7.183

*R*_wp_: weight-profile goodness of fit value.

**Table 3 materials-13-05006-t003:** Mixture design of magnesium oxysulfate cement pastes.

Sample No.	Mixture Design (wt.%)				
	Steel Slag	Caustic Calcined Magnesia	MgSO_4_·7H_2_O	Water	Citric Acid
Control	0	100	50	50	0.5
SS10	10	90	45	50	0.5
SS20	20	80	40	50	0.5
SS30	30	70	35	50	0.5
SS40	40	60	30	50	0.5
SS60	60	40	20	50	0.5

**Table 4 materials-13-05006-t004:** Component concentrations in the control, SS10, SS20, and SS30 with CO_2_ treatment curing at 3 days, 28 days, and after immersed in water for 28 days.

Sample	Period (days)	Phase Content (wt.%)										*R*_wp_(%)
		517 Phase	Periclase	Brucite	Magnesite	Quartz	Calcite	Gypsum	C_2_S	C_4_AF	ACn	
Control	3	19.8	28.1	5.3	5.6	0.9	-	-	-	-	40.3	7.079
	28	20.1	13.3	15.7	6.3	0.8	-	-	-	-	43.8	7.568
	28^a^	22.9	7.7	18.9	6.1	1.1	-	-	-	-	43.3	8.598
SS10	3	13.6	22.1	2.4	8.8	0.7	0.7	1.5	2.1	-	48.1	7.134
	28	15.9	15.1	8.7	8.1	0.6	1.1	1.9	0.5	-	48.1	6.994
	28^a^	15.6	7.9	17.0	6.7	0.4	0.5	2.1	0.1	-	49.7	9.164
SS20	3	0.3	26.1	6.6	7.6	0.8	1.3	1.7	4.8	6.3	44.5	7.129
	28	0.5	20.2	10.7	7.1	1.1	1.6	3.1	2.3	2.3	51.1	8.567
	28^a^	3.7	19.1	13.1	7.2	0.7	0.6	2.7	0.9	2.1	49.9	8.131
SS30	3	-	24.1	5.7	6.2	0.6	0.9	3.2	9.3	5.1	44.9	7.213
	28	0.5	18.8	7.8	6.7	0.3	1.2	5.2	3.2	3.3	53.0	7.466
	28 ^a^	1.1	18.7	8.7	5.6	0.5	1.4	5.1	2.3	1.1	55.3	8.586

^a^ Period of water curing after air curing for 28 days.

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
