# Peer review of "Effect of Carbonation on the Water Resistance of Steel Slag—Magnesium Oxysulfate (MOS) Cement Blends"

_materials, 2020, doi:10.3390/ma13215006_

Round 1
Reviewer 1 Report
This is an appropriate investigation, focused on the application of steel slag in the manufacture of magnesium oxysulfate (MOS) cement. An adequate introduction is included but the objective and research design should be expanded in the last paragraph, giving more details of the design that is applied.
A very reduced materials section is included, the particle size distribution curves should be incorporated using laser granulometry equipment, and density, water absorption capacity, and other characterization tests should be included.
Only four mixtures were manufactured, and in my opinion these mixtures are not enough to obtain relevant conclusions.
It is necessary to describe better how the mortars were made, and it must be explained which international standard was follow to manufacture these mixtures. Also, the dosages of each mixture should be included.
Lastly, the conclusions section is very reduced and poorly structured. A more extended introduction to the conclusions should be included in the first part, and later to divide into 6 to 8 specific conclusions, and finally a general conclusion should be presented at the end of this section.
Author Response
Thank you for your letter and for the reviewers’ comments concerning our manuscript entitled “Effect of carbonation on the water resistance of steel slag-magnesium oxysulfate (MOS) cement blends”. Those comments are all valuable and very helpful for revising and improving our paper, as well as the important guiding significance to our researches. We have studied comments carefully and have made correction which we hope meet with approval.
Point 1: An adequate introduction is included but the objective and research design should be expanded in the last paragraph, giving more details of the design that is applied.
Response 1: Considering the reviewer’s suggestion, we have given more details of the design in the last paragraph in the introduction part (Lines 74-78, page 1).
Point 2: A very reduced materials section is included, the particle size distribution curves should be incorporated using laser granulometry equipment, and density, water absorption capacity, and other characterization tests should be included.
Response 2: The particle size distribution curves have been shown in Figure 1 (Line 92, page 3), more details of physical properties are shown in table 1. i. e. density, water absorption and specific surface area (Line 95, page 3).
Point 3: Only four mixtures were manufactured, and in my opinion these mixtures are not enough to obtain relevant conclusions.
Response 3: In response to reviewer’s requests, we supplemented the data for the other two mixtures in Figure 2 and Figure 3 (Lines 164 and 168, page 7). The compressive strength before and after carbonation of the samples with steel slag content of 40wt.% and 60wt.% were added. When the steel slag content is greater than 30wt.%, the strength is not high enough to meet the engineering requirements (Lines 159-162, page 5), so the following data was not shown in the previous paper, leading to the lack of rigor of the paper. And further researches will be studied in the next stage.
Point 4: It is necessary to describe better how the mortars were made, and it must be explained which international standard was follow to manufacture these mixtures. Also, the dosages of each mixture should be included.
Response 4: We are very sorry for our negligence of the lack of details for preparation. The dosages of each mixture are supplemented in Table 3, and the preparation process is described in detail, according to JC/T 729-2005. First, the mixture was stirred at a low speed rate of 60 r/min for 120 s, then the stirring was stopped for 15 s, and then the mixture was stirred at a high rate of 300 r/min for 120 s (Lines 100-108, page 4).
Point 5: Lastly, the conclusions section is very reduced and poorly structured. A more extended introduction to the conclusions should be included in the first part, and later to divide into 6 to 8 specific conclusions, and finally a general conclusion should be presented at the end of this section.
Response 5:
We have re-written this part according to the suggestion.
The structure of the conclusion was readjusted, the conclusion was expanded to six, the beginning part was supplemented, and a general conclusion was made at the end (Lines 317-356, pages 18-19).

Reviewer 2 Report
This in an interesting paper dealing with the combined use of CO2 treatment and steel slag for the production of water resistant MOS-based mixtures.
The topic is novel and the paper is well structured, references are adequate and discussion of results is reasonable.
The authors should be underline the possible use of carbon dioxide treatment and steel slag in the industrial sector.
At the same time, no information concerning the environmental advantages deriving from the use of industrial by products (less CO2 emission, less energy demand and less natural raw materials consumption due to the partial substitution of cement) are present within the text. Moreover, the evaluation of the reduction of CO2 emission related to the CO2 treatment is totally missing. Several simplified formulas are reported in the recent scientific literature i.e.
Damineli et al., 2010, Measuring the eco-efficiency of cement use Cement Concr. Compos., 32 (8) (2010), pp. 555-562
Gettu et al., 2018 Considerations of sustainability in the mixture proportioning of concrete for strength and durability Spec. Publ., 326 (2018), pp. 5.1-5.10
Coppola et al., 2019 An Empathetic Added Sustainability Index (EASI) for cementitious based construction materials J. Cle. Pro., 220 (2019), pp. 475-482
Other minor comments:
- Please, revise the format of table 2 and 4
- Define Rwp
- Figures 7-8-9 are of poor quality
- SEM images are useless in this resolution. Please, upload high-res images
- Authors contributions are missing
- Check references. between 26 and 27 is present a not numbered reference
Author Response
Thank you for your letter and for the reviewers’ comments concerning our manuscript entitled “Effect of carbonation on the water resistance of steel slag—magnesium oxysulfate (MOS) cement blends”. Those comments are all valuable and very helpful for revising and improving our paper, as well as the important guiding significance to our researches. We have studied comments carefully and have made correction which we hope meet with approval.
Point 1: The authors should be underline the possible use of carbon dioxide treatment and steel slag in the industrial sector. At the same time, no information concerning the environmental advantages deriving from the use of industrial by products (less CO2 emission, less energy demand and less natural raw materials consumption due to the partial substitution of cement) are present within the text. Moreover, the evaluation of the reduction of CO2 emission related to the CO2 treatment is totally missing.
Response 1: As reviewer suggested that, the first part of the introduction is supplemented with environmental advantages deriving from the use of industrial by steel slag (Lines 26-35, page 1). The references provided are cited in the article (Lines 362-373, page 19).
Global warming induced by increased concentrations of CO2 emissions by human activities has been the largest environmental threat of the 21st century and will become more severe. Portland cement, a major civil engineering material, emits more than 4 billion tons of CO2 annually, accounting for 5-10% of global emissions. Therefore, it is urgent to find an alternative to Portland cement to achieve sustainable development and reduce the greenhouse effect.
Magnesium oxysulfate (MOS) cement is a green and environment-friendly civil engineering material prepared using caustic calcined magnesia and an aqueous solution of magnesium sulfate. Since the calcination temperature of caustic calcined magnesia is lower than that of Portland cement (~ 800 vs. 1450 oC), the carbon dioxide emitted by MOS cement is only 40-50% of that of PC.
Point 2:
Please, revise the format of table 2 and 4
Define Rwp
Figures 7-8-9 are of poor quality
SEM images are useless in this resolution. Please, upload high-res images
Authors contributions are missing
Check references. between 26 and 27 is present a not numbered reference
Response 2:
We have supplemented and modified according to the suggestions.
The issues in Table 2 and 4 have been corrected .
Rwp has been defined in Table 2 (Line 98, page 3).
Figure 7-8-9 was modified to remove some unnecessary scale lines and adjust the thickness of the auxiliary lines (Line 223, page 10; Line 228, page 12; Line 257, page 13).
The resolution of SEM images was adjusted from 400×533 to 768×1024 (Line 271, page 15; Line 282, page 16).
Authors contributions have been supplemented (Lines 357-361, page 19).
The references have been revised.

Round 2
Reviewer 1 Report
In my opinion the authors have made the changes proposed by the reviewers, and this paper could be accepted for publishing